# FormaRL: Enhancing Autoformalization with no Labeled Data

**Yanxing Huang[1], Xinling Jin[4], Sijie Liang[5], Peng Li[3]\*, Yang Liu[2,3]\***
[1]Department of Mathematics Sciences, Tsinghua University
[2]Dept. of Comp. Sci. & Tech., Institute for AI, Tsinghua University, Beijing, China
[3]Institute for AI Industry Research (AIR), Tsinghua University
[4]School Of Computer Science And Technology, Tongji University
[5]Faculty of Science Mathematics and applied mathematics, Beijing Forestry University

## Abstract

Autoformalization is one of the central tasks in formal verification, while its advancement remains hindered due to the data scarcity and the absence efficient methods. In this work we propose **FormaRL**, a simple yet efficient reinforcement learning framework for autoformalization which only requires a small amount of unlabeled data. FormaRL integrates syntax check from Lean compiler and consistency check from large language model to calculate the reward, and adopts GRPO algorithm to update the formalizer. We also curated a proof problem dataset from undergraduate-level math materials, named **uproof**, in the hope to facilitate the exploration of autoformalization and theorem proving in advanced math. Experiments show that FormaRL can increase the pass@1 autoformalization accuracy of Qwen2.5-Coder-7B-Instruct by $4 \sim 6x$ ($4.04\% \rightarrow 26.15\%$ on ProofNet and $2.4\% \rightarrow 9.6\%$ on uproof) with merely 859 unlabeled data. And on uproof our method also achieved a strong improvement in out-of-distribution performance compared to existing open-source state-of-the-art autoformalizers on both pass@1 accuracy ($6.2\% \rightarrow 9.6\%$) and pass@16 accuracy ($24.4\% \rightarrow 33.6\%$). Training code of FormaRL is open-sourced at https://github.com/THUNLP-MT/FormaRL.

## 1 Introduction

Mathematical reasoning has long been regarded as a cornerstone of scientific and technological advancement. Recent large language models (LLMs) have made advancements in solving math problems (OpenAI et al., 2024a; Qwen et al., 2024; Yang et al., 2024a). Large reasoning models (LRMs) gain stronger reasoning abilities, and achieved impressive results on competition level math problems (OpenAI et al., 2024b; DeepSeek-AI et al., 2025a; Team et al., 2025).

Although language models excel in many mathematical tasks, they still struggle with theorem proving. An important reason is that theorem proving typically requires formal verification, and one of the critical steps of it, autoformalization, remains challenging for models (Li et al., 2024b; Yang et al., 2024b). Autoformalization is the process which translates natural language mathematics into formal languages (Li et al., 2024b; Yang et al., 2024b), such as Lean (Moura & Ullrich, 2021), Isabelle (Tobias Nipkow, 2002), or Coq (Yves Bertot, 2013). It is the start point of formal verification for a theorem. And it is crucial for creation of large scale training dataset of following steps of theorem proving (Wu et al., 2024; Li et al., 2024a; Lin et al., 2025; Xin et al., 2025; Castelvecchi, 2024; Xin et al., 2024a;b).

Existing models demonstrate certain ability on autoformalization of elementary mathematics, but their performance significantly drops when it comes to that of advanced mathematics (see Table 4). Advanced math covers a much wider range of concepts and is much more

---

*Corresponding authors: Peng Li (lipeng@air.tsinghua.edu.cn) and Yang Liu (liuyang2011@tsinghua.edu.cn)

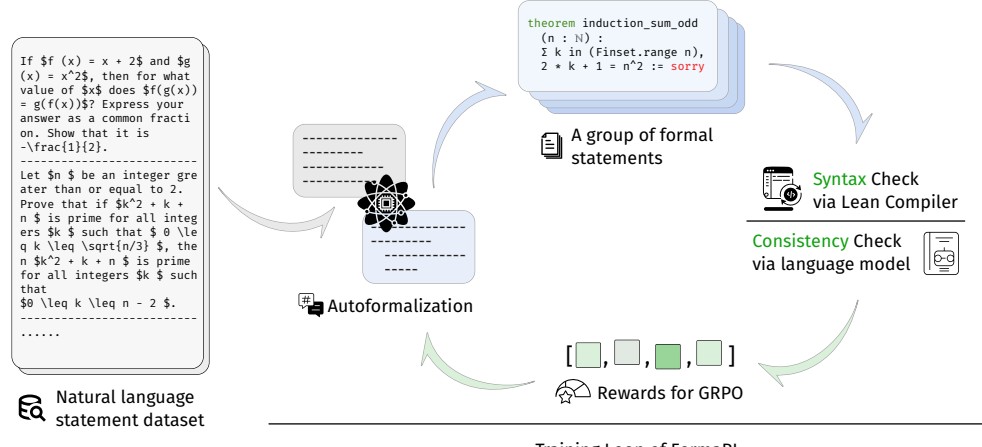

Figure 1: Illustration of FormaRL training loop. We combined lean syntax check from the compiler and LLM based semantic check to assess the quality of formalizations, and adopted GPRO algorithm to update our formalizer.

complex than contest level math, thus is harder to formalize (Azerbayev et al., 2023,). The lack of training data further limits autoformalization performance in this area. Some prior works (Xin et al., 2024a;b; Li et al., 2024a; Lin et al., 2025; Xin et al., 2025) train autoformalizer on existing large scale formalization datasets, such as MMA (Jiang et al.), Lean Workbook (Ying et al., 2024) or some in-house datasets. Other works boost model performance via supervised fine-tuning (SFT) on manually annotated data (Ying et al., 2024) or informalized statement pairs (Liu et al., 2025; Jiang et al.). However, due to the high cost of human annotation and the inefficiency of training recipe, their performance remains limited.

For the above challenge, we propose a method called **FormaRL**. We utilize reinforcement learning to train our model, with rewards from Lean compiler and LLMs, as shown in Figure 1. Our method is more effective and efficient compared to previous SFT methods. We use training data which is unlabeled, and only 1% of the amount compared to prior works, to reach superior performance. Additionally, our method is applicable to various models, including mathematical foundational models and existing autoformalizers. we also create a dataset, named **uproof**, to evaluate out-of-distribution autoformalization performance in advanced math. It contains 5,273 proof problems from 14 classical textbooks, which covers a wide range of topics from undergraduate-level math.

In summary, our contributions are three-folded:

- We propose **FormaRL**, a simple yet effective RL based training framework to enhance model ability of autoformalization with far less training data.

- We create a benchmark named **uproof**, bridging the gap of evaluation of out-of-distribution autoformalization for advanced math problems.

- We conduct extensive experiments, and find that existing models fall short in advanced math autoformalization, while our proposed method reaches promising performance especially on advanced math autoformalization. Our ablation study further guarantees the effectiveness of FormaRL.

## 2 Related work

### 2.1 Autoformalization

Autoformalization aims to automatically translate mathematical materials in natural language into machine-verifiable formal code (Li et al., 2024b). Broadly speaking, it include the translation of both the statements and proofs of a math problem (Cunningham et al., 2023; Jiang et al., 2023; Zhao et al., 2023; Murphy et al., 2024; Patel et al.), while another line of works primarily focus on the translation of the problem statements (Wu et al., 2022; Jiang et al.; Azerbayev et al., 2023,).

Autoformalization is a challenging task especially for currently prevalent data-driven approaches (Li et al., 2024b). To alleviate the scarcity of informal-formal corpora, researchers adopted various methods to synthesize large scale datasets for training. This LLM based informalization (Azerbayev et al., 2023,; Jiang et al.; Liu et al., 2025), or utilizing in-context learning (ICL) capability to create an expert iteration pipeline for autoformalization (Wu et al., 2022; Ying et al., 2024). One major difference in autoformalization compared to machine translation is the existence of formal verifier. They can provide accurate feedback on the accuracy of the formalized statements and proofs. The verifier is widely used to conduct rejection sampling (Poiroux et al., 2024) or expert iteration (Jiang et al., 2023; Murphy et al., 2024) to enhance autoformalization. It can also contribute to constructing objective evaluation metrics (Azerbayev et al., 2023,; Ying et al., 2024; Liu et al., 2025).

### 2.2 Reinforcement learning

Reinforcement learning (RL) is a machine learning paradigm where an agent learns to make decisions by interacting with the environment to maximize cumulative rewards. Unlike supervised learning, which relies on large scale labeled datasets, RL algorithms learn through trial and error, guided by a reward signal that evaluates the quality of actions taken.

In the development of modern LLMs, there is many works that leverage RL algorithms to optimize their performance. Reinforcement learning from human feedback (RLHF) was proposed to align LLMs behaviour with human preferences and values (Ouyang et al., 2022; Ziegler et al., 2020). Recent work demonstrates RL's effectiveness in both informal math problem solving (Lightman et al., 2023; Shao et al., 2024) and formal theorem proving (Xin et al., 2024b; Lin et al., 2025; Xin et al., 2025). RL is also the key to the paradigm shift from LLM to LRM (DeepSeek-AI et al., 2025a; Team et al., 2025). RL-based algorithm surpasses SFT is both data efficiency and final performance according to some recent studies (Zeng et al., 2025; Li et al., 2025; Yu et al., 2025).

## 3 Preliminaries

In this work we primarily focus on formal language Lean 4, with particular attention to the translation of mathematical problem statements. In Lean 4, it is possible to simulate the completion of a proof using the keyword "sorry". If a theorem statement is syntactically correct and either properly proved or concluded with "sorry", the Lean compiler returns a "no goals" message, indicating that the statement is accepted.

A typical problem statement in Lean 4 is written as follows:

```
theorem exercise_1_13a {f : ℂ → ℂ} ( Ω : Set ℂ ) (a b : Ω) (h : IsOpen
    Ω) (hf : DifferentiableOn ℂ f Ω) (hc : ∃ (c : ℝ), ∀ z ∈ Ω, (f
    z).re = c) : f a = f b := sorry
```

The "theorem" keyword is used to declare a named theorem, while "example" can be used for unnamed statements. Each declaration is concluded with " s̄orry" if a proof is omitted. Here we also provide some descriptions of each dataset involved in our experiments.

- miniF2F (Zheng et al., 2022) is a formal theorem proving benchmark proposed by OpenAI. It consists 488 problems from elementary math with several variants in

Lean, Coq and Isabelle. The original purpose of miniF2F was to create a universal benchmark across different formal languages. The problems in miniF2F are drawn from high-school exercises and contests such as AIME, AMC and the IMO. In our experiments we use the version from Xin et al. (2024a) as an autoformalization benchmark for elementary math.

- ProofNet (Azerbayev et al., 2023,) was proposed both as a theorem proving benchmark and an autoformalization benchmark. It primarily focus on undergraduate-level math, manually collected and translated 371 problems into Lean. It covers a wide range of topics from advanced mathematics, from real and complex analysis to algebra and topology. Again we use the version provided by Xin et al. (2024a) as an autoformalization benchmark for advanced math.

- Lean Workbook (Ying et al., 2024) is a large scale Lean 4 problem set formalized from contest level math problems in natural language. They crawled the raw problems from AOPS and translated them into Lean 4 using a formalizer trained by themselves. After filtering these results they ended up with 25.2k Lean 4 translation pairs in total, and most of them also belong to elementary math. We primarily use this dataset to train formalizers via SFT as our baseline. Recent researches showed that formalizers trained on this dataset already exhibits strong performance compared to other formalization corpora (Liu et al., 2025).

## 4 Methodology

We begin our experiments by designing a reward to assess the translation, which is then integrated into a reinforcement learning framework to iteratively refine translation performance. Our reward design eliminates the need for annotated translation datasets, enabling training of a formalizer without reliance on manual demonstrations.

### 4.1 Reward design

We adapted the data filtering method in Lean Workbook (Ying et al., 2024) as the core of our reward system. This process involves two sequential validation stages:

- **Syntax Check (SC)**: Extracted translations will firstly undergo automated validation via the Lean 4 compiler to ensure syntactic correctness. This ensures that the outputs are valid lean 4 code.

- **Consistency Check (CC)**: After stripping comments and metadata, the translation's semantic alignment with the original problem is evaluated using a large language model (LLM).

For each response from the translation model, a reward of "1.0" is assigned only when it passed both SC and CC, otherwise the reward will always be "0.0". This hybrid rule-based and LLM-driven assessment ensures robustness, as empirical results demonstrate that neither component alone suffices for reliable validation, see section 5.3 for more details

### 4.2 Training method

We adopted a simplified version of Group Relative Policy Optimization (GRPO) as our training algorithm. This is also the algorithm behind the success of DeepSeek-R1 (Shao et al., 2024; DeepSeek-AI et al., 2025a). For each question $q$, GRPO algorithm samples a group of outputs $\{o_1, o_2, \ldots, o_G\}$ from the old policy model $\pi_{\theta_{old}}$, and then optimize the policy model by maximizing the following objective:

$$\mathcal{J}_{GRPO}(\theta) = \mathbb{E}[q \sim P(Q), \{o_i\}_{i=1}^{G} \sim \pi_{\theta_{old}}]$$

$$\frac{1}{G}\sum_{i=1}^{G}\frac{1}{|o_i|}\sum_{t=1}^{|o_i|}\left\{\min\left[\frac{\pi_\theta(o_{i,t}|q,o_{i,<t})}{\pi_{\theta_{old}}(o_{i,t}|q,o_{i,<t})}\hat{A}_{i,t}, \text{clip}(\frac{\pi_\theta(o_{i,t}|q,o_{i,<t})}{\pi_{\theta_{old}}(o_{i,t}|q,o_{i,<t})}\hat{A}_{i,t}, 1-\epsilon, 1+\epsilon)\right]\right\}$$

Here $\epsilon$ and $\beta$ are hyperparameters, and $\hat{A}_{i,t}$ is the advantage calculated based on relative rewards of the outputs in each group. More accurately it is $\hat{A}_{i,t} = \tilde{r}_i = \frac{r_i - \text{mean}(\mathbf{r})}{\text{std}(r)}$, where $\mathbf{r}$ is the observed reward.

A notable departure from standard GRPO is the omission of the KL divergence regularization term. This simplification, supported by recent empirical studies (Meng et al., 2024; Yu et al., 2025), maintains training stability while reducing computational overhead, and can lead to better performance in many cases.

### 4.3 Quality assessment

While Lean 4's syntax checker provides rigorous formal validation, the reliability of the consistency check (CC) warrants closer examination. Inaccuracies in this process could lead to reward hacking or degrade the overall effectiveness of training.

We frame the evaluation of CC as a binary classification task. To assess recall, we applied the consistency check to samples from existing formal datasets paired with their ground truth translations, measuring the rate at which correct samples were accepted. To estimate specificity, we prompted a large language model to deliberately generate incorrect translations by modifying original statements, such as adding, removing, or altering conditions or conclusions, and then tested whether the CC could successfully reject these perturbed examples.

Unless otherwise noted, DeepSeek-V3 (DeepSeek-AI et al., 2025b) was used for CC during most of the FormaRL training process, while Qwen2.5-7B-Instruct (Qwen et al., 2024) served as the evaluation model. The performance of the consistency check is summarized in Table 1.

| Model | miniF2F | | ProofNet | |
|---|---|---|---|---|
| | Recall ↑ | Specificity ↑ | Recall ↑ | Specificity ↑ |
| Qwen2.5-7B-Instruct | 87.29% | 92.22% | 78.59% | 79.68% |
| DeepSeek-V3 | 88.52% | 98.57% | 76.82% | 93.53% |
| GPT-4o | 84.22% | 97.95% | 72.51% | 90.84% |

Table 1: Performance of the consistency check across different models and datasets.

These results indicate that the consistency check performs reasonably well on simpler datasets like miniF2F, but struggles more with complex mathematical content, as seen in ProofNet, where both recall and specificity are lower. This suggests that using CC as a filtering or selection mechanism may be problematic, particularly in high-throughput scenarios like pass@8 sampling, where specificity can drop sharply (e.g., to 52.31% for miniF2F and 16.25% for ProofNet), allowing a significant number of incorrect samples into the generated dataset.

To mitigate this, our subsequent experiments rely solely on the syntax check (SC) for selection. For each problem, only the first candidate that passes SC proceeds to consistency checking and contributes to the final pass rate.

## 5 Experiments

### 5.1 Dataset

To construct our dataset, we curated 14 classical mathematics textbooks spanning core undergraduate curricula, including mathematical analysis, linear algebra, abstract algebra, real analysis, complex analysis, functional analysis, topology, probability, statistics and commutative algebra.

The raw textbook content was preprocessed by converting PDF files into markdown format, followed by segmentation into smaller chunks of reasonable length. We required GPT-4o to extract the lemmas, theorems and exercises from these documents. After that we required GPT-4o again to filter out incomplete problems and reformat math formulas into standard LaTeX syntax.

This results in uproof, a large scale dataset of undergraduate-level proof problems. Here we list some examples in uproof in Table 2.

| Problem | Category |
|---|---|
| If $F : D \to P$ is a conformal map from the disc $D$ to a polygonal region $P$, show that $F$ extends to a continuous bijection from the closure of $D$ to the closure of $P$. | Analysis |
| Let $\alpha = (2 + \sqrt{2})(3 + \sqrt{3})$ and consider the extension $E = \mathbb{Q}(\alpha)$. Show that $\alpha$ is not a square in $F = \mathbb{Q}(\sqrt{2}, \sqrt{3})$ | Algebra |
| Prove that every closed nonorientable surface has a 2-sheeted orientable covering space. | Topology |
| Show that if $\frac{\sqrt{n}(X_n - \mu)}{\sigma}$ converges in distribution to $N(0,1)$, then the expression $\frac{\sqrt{n}(\bar{X}_n - \mu)}{S_n}$ converges in distribution to $N(0,1)$ as well. | Probability |

Table 2: Samples from uproof dataset.

## 5.2 Results

We utilize two well-known datasets in formal theorem proving, miniF2F and ProofNet, for training with FormaRL. It is important to note that these two datasets contains merely 859 statements in total, which is significantly smaller than the data requirement of existing SFT-based methods (25.2k in Lean Workbook (Ying et al., 2024) and 243k for RAutoformalizer Liu et al. (2025)). The ground truth translations in these datasets are not involved in FormaRL either. Nevertheless, our experiments demonstrate that FormaRL already exhibits superior performance in autoformalization.

In our experiments we used Lean Workbook as the training dataset for our baseline formalizers via SFT. It is worth mentioning that this dataset was synthesized using both miniF2F and ProofNet, making these benchmarks in-distribution for all formalizers evaluated. In our main experiment, we tested two different base models, Qwen2.5-Coder-7B-Instruct (Hui et al., 2024) and DeepSeek-Math-7B (Shao et al., 2024). We randomly selected 1,000 elements from uproof dataset as our validation split. Since our primarily focus is on the generalization ability of different strategies, the results on uProof are particularly informative, as they reflect the models' out-of-distribution performance.

Our main experiment results are summarized in Table 3.

We can see that FormaRL outperformed SFT baselines by a large margin, demonstrating its strong generalization capability.

DeepSeek-Math-7B-Instruct performs poorly out of the box and struggles to generate correct formalizations without fine-tuning, thereby would severely impact the efficiency of RL. So we applied a minimal warm-up for this model, specifically, we randomly selected 1k translation pairs from Lean Workbook and trained DeepSeek-Math-7B-Instruct for 1 epoch. This does not introduce excessive additional training or data requirement. Meanwhile, Qwen2.5-Coder-7B-Instruct is trained via FormaRL entirely from scratch.

We also evaluated the effect of applying FormaRL on top of a pre-trained formalizer. While this setup showed that FormaRL can further improve performance after extensive SFT, the improvement plateaued, suggesting a limited upper bound. Notably, the most substantial performance gains were observed when using FormaRL without any prior SFT.

| Model | Method | SC Pass Rate | Final Pass Rate | Δ |
|---|---|---|---|---|
| **Pass@1** | | | | |
| DeepSeek-V3 | ICL | 1.3% | 1.1% | |
| GPT-4o | | 2.9% | 2.2% | |
| DeepSeek-Math-7B-Instruct | | 0.2% | 0.1% | |
| Qwen2.5-Coder-7B-Instruct | | 3.7% | 2.4% | |
| RAutoformalizer (Liu et al., 2025) | - | 14.1% | 6.2% | |
| | FormaRL | 20.1% | **11.9**% | + 5.7% |
| DeepSeek-Math-7B-Instruct | SFT | 21.2% | 7.8% | |
| | FormaRL | 14.4% | 8.6% | + 0.8% |
| Qwen2.5-Coder-7B-Instruct | SFT | 20.4% | 7.5% | |
| | FormaRL | 18.6% | 9.6% | + 2.1% |
| **Pass@8** | | | | |
| RAutoformalizer (Liu et al., 2025) | - | 48.0% | 20.0% | |
| | FormaRL | 44.8% | 21.9% | + 1.9% |
| DeepSeek-Math-7B-Instruct | SFT | 54.7% | 17.0% | |
| | FormaRL | 46.1% | 22.8% | + 5.8% |
| Qwen2.5-Coder-7B-Instruct | SFT | 52.3% | 15.8% | |
| | FormaRL | 62.2% | **29.8**% | +14.0% |
| **Pass@16** | | | | |
| RAutoformalizer (Liu et al., 2025) | - | 65.2% | 24.4% | |
| | FormaRL | 53.4% | 25.7% | + 1.3% |
| DeepSeek-Math-7B-Instruct | SFT | 66.4% | 22.5% | |
| | FormaRL | 57.7% | 27.2% | + 4.7% |
| Qwen2.5-Coder-7B-Instruct | SFT | 63.6% | 21.2% | |
| | FormaRL | 66.7% | **33.6**% | +12.4% |

Table 3: The out-of-distribution accuracy of autoformalization on uproof split (advanced math). ICL is for "in-context learning". The SFT approach utilizes 25.2k Lean Workbook dataset for training, while our FormaRL approach only uses 859 unlabeled statements from miniF2F and ProofNet. As RAutoformalizer has been already fine-tuned on a 243k dataset for autoformalization, whose scale is much larger than our SFT dataset, we do not conduct further SFT on it.

To assess in-distribution performance, we evaluated some of these models on miniF2F and ProofNet, relevant results are listed in Table 4. While these benchmarks are less central to our primary focus, they serve as useful supplementary references.

Both propriaty models, DeepSeek-V3 and GPT-4o exhibit poor performance especially in advanced math such as ProofNet dataset. This mainly stems from their low SC pass rate. More detailed examinations on this can be found in Appendix A.1.

## 5.3 Ablation study

Our experiments also suggest that FormaRL is a minimal possible RL framework for autoformalization. That means if we eliminate any component in the reward calculation, the RL training would cause severe reward hack and no meaningful results. And at the same time, we can observe consistent performance increment when integrating both SC and CC check.

If we calculate the reward without CC check, the model will quickly learn to generate the same simple statement that is not relevant to the given problem so that it can pass all syntax checks. On the other hand if we eliminate SC check, the model will quickly learn to include natural language statements from the original problem in its response, preserving good consistency but that is not formalization. We list some typical cases and evaluation results in Table 5

| Model | Method | miniF2F | ProofNet |
|---|---|---|---|
| DeepSeek-V3
GPT-4o | ICL | 24.59% / 20.08%
47.95% / **38.93%** | 2.70% / 2.70%
4.04% / **3.23%** |
| RAutoformalizer (Liu et al., 2025) | -
FormaRL | 49.18% / 25.00%
83.20% / **53.89%** | 28.30% / 19.68%
58.49% / **46.90%** |
| DeepSeek-Math-7B-Instruct | SFT
FormaRL | 82.17% / 48.77%
86.27% / **59.63%** | 35.04% / 22.64%
42.59% / **31.81%** |
| Qwen2.5-Coder-7B-Instruct | SFT
FormaRL | 88.32% / 56.35%
81.56% / **57.58%** | 35.85% / 18.87%
35.58% / **26.15%** |

Table 4: The pass@1 accuracy of in-distribution autoformalization performance. ICL is for "in-context learning". We use miniF2F dataset to test the autoformalization performance of elementary math, and ProofNet for advanced math. Similar to Table 3, for each test, we report the SC pass rate and the final pass rate after SC and CC.

| Method | Case | Pass Rate | |
|---|---|---|---|
| w/o SC | holomorphic_open_omega f Ω → Re_constant f → Constant f := sorry | SC
CC
SC&CC | 0.27%
74.66%
0.00% |
| w/o CC | theorem realPartConstantImpliesConstant: False := sorry | SC
CC
SC&CC | 100%
0%
0% |
| CC & SC | theorem prove_f_is_constant {f : ℂ → ℂ} (h_f_holomorphic : ∀ (z : ℂ), ∀ (U : Set ℂ), IsOpen U ∧ ∀ (w : ℂ), w ∈ U → ∃ (L : ℂ), ∀ (z1 z2 : ℂ), z1 ∈ U ∧ z2 ∈ U → (f z1 − f z2 = L * (z1 − z2))) (h_im_constant : ∀ (z : ℂ), ∀ (w : ℂ), f z = f w → Im (f z) = Im (f w)) : ∀ (z1 z2 : ℂ), f z1 = f z2 := sorry | SC

CC

SC&CC | 35.58%

57.14%

26.15% |

Table 5: Ablation study on reward design. These formalizers are trained from Qwen2.5-7B-Instruct and tested on ProofNet. These are formalizations from the same proof problem in complex analysis: "Suppose that $f$ is holomorphic in an open set $\Omega$. Prove that if $\text{Im}(f)$ is constant, then $f$ is constant".

The performance of FormaRL heavily relies on the distinguishing ability of the LLM used in CC. We also tried some weaker language models in the training process. By switching the backend of CC from DeepSeek-V3 to Qwen2.5-7B-Instruct, we can observe performance drop on all benchmarks under pass@8 or pass@16 settings. Typically models trained with weaker LLM in FormaRL exhibit weaker performance in keeping the consistency in autoformalization, but tend to have higher SC pass rate in sampling. This limits their potential to gain improvement from multiple sampling. Nevertheless, these models still outperformed SFT baselines, with significantly less data requirement. Relevant results are summarized in Table 6.

A formal statement that passes both SC and CC checks is typically a promising formalization, however, since we used a large language model to conduct consistency check in both training and evaluation, there exists the possibility of reward hack. We have selected 100 samples for both formalizer, and done some manual reviews on samples generated by formalizer trained via FormaRL and RAutoformalizer. As shown in Table 7, since the formalizer trained with RL did not exhibit significantly higher CC pass rate under LLM-based review than manual, There is no evidence of reward hack in our experiments. We also did some ablation study on the LLM backend of CC, relevant results can be found in Table 10 in the appendix. These results also support the non-existence of reward hack after FormaRL.

| Method | Base Model | Settings | Performance |
|---|---|---|---|
| SFT | Qwen2.5-Coder-7B-Instruct | pass@1 | 20.4% / 7.5% |
| FormaRL(w/ deepseek) | Qwen2.5-Coder-7B-Instruct | pass@1 | 18.6% / 9.6% |
| FormaRL(w/ qwen) | Qwen2.5-Coder-7B-Instruct | pass@1 | 28.5% / **11.7%** |
| SFT | Qwen2.5-Coder-7B-Instruct | pass@8 | 52.3% / 15.8% |
| FormaRL(w/ deepseek) | Qwen2.5-Coder-7B-Instruct | pass@8 | 62.2% / **29.8%** |
| FormaRL(w/ qwen) | Qwen2.5-Coder-7B-Instruct | pass@8 | 69.8% / 27.6% |
| SFT | Qwen2.5-Coder-7B-Instruct | pass@16 | 63.6% / 21.2% |
| FormaRL(w/ deepseek) | Qwen2.5-Coder-7B-Instruct | pass@16 | 66.7% / **33.6%** |
| FormaRL(w/ qwen) | Qwen2.5-Coder-7B-Instruct | pass@16 | 79.8% / 31.3% |

Table 6: Ablation studies on the CC reward in training. We can see that models trained with DeepSeek-V3 is consistantly more performant than that with Qwen2.5-7B-Instruct. Performance is evaluated on uproof dataset and reported as SC pass rate / final pass rate.

| Formalizer | Manual | Qwen2.5-7B-Instruct |
|---|---|---|
| RAutoformalizer(Liu et al., 2025) | 25.0% | 37.4% |
| Qwen2.5-Coder-7B-Instruct with FormaRL | 38.8% | 50.4% |

Table 7: Comparesion of CC accuracy after SC under both manual review and LLM-based review.

# 6 Conclusion and future work

FormaRL is a simple RL based training framework for autoformalization. It requires only a small number of unlabeled training data but still exhibits superior performance compared to vanilla SFT. Recently there are some more advanced methods proposed to enhance the evaluation or sampling process for autoformalized problems, such as Bidirectional Extended Definitional Equivalence (Liu et al., 2025) and dependency retrieval augmentation (Liu et al., 2025). They also exhibit great improvement either in evaluation precision or autoformalization quality. These methods are easy to integrate into our training pipeline. We can add the contents provided by dependency retrieval to the prompts during training and sampling, and use BEq in reward design. We believe this will bring the performance of autoformalization to the next level.

Theorem proving is the central task of modern math research, thus is of more importance than math contests. Some recent experiments suggest that the data amount required to train a powerful LRM is relatively low (Li et al., 2025; Zeng et al., 2025), so maybe we are already able to train a strong theorem prover with a proper RL algorithm and formal verification on our uproof dataset or NaturalProofs (Welleck et al., 2021). We are optimistic about future progress in theorem proving especially in advanced math, an area that remains largely unexplored.

# Acknowledgement

This work is supported by the National Natural Science Foundation of China (No. 62276152), and funding from Wuxi Research Institute of Applied Technologies, Tsinghua University under Grant 20242001120.

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

# A Appendix

## A.1 Case studies

We observed that both large scale LLMs, GPT-4o and DeepSeek-V3, can not exhibit promising performance on autoformalization according to our experiments. Things get even worse

when it comes to advanced math. For example on ProofNet, their syntax check pass rate is both below 5%. By examine the cases generated by GPT-4o we can figure out the reason behind this.

Here is a sample of response from GPT-4o, and the original problem comes from ProofNet dataset:

```
To translate this natural language problem into a theorem in Lean4, we
    first need to introduce the necessary concepts, such as holomorphic
    functions and the real part of a complex function. Assuming we have
    these concepts defined, the theorem can be captured as follows:

'''lean
theorem holomorphic_const_real_part_is_const {f : ℂ → ℂ} {Ω : set ℂ}
  (hΩ : is_open Ω) (hf : ∀ z ∈ Ω, differentiable_at ℂ f z)
  (h_re_const : ∃ c : ℝ, ∀ z ∈ Ω, (f z).re = c) :
  ∀ z1 z2, z1 ∈ Ω → z2 ∈ Ω → f z1 = f z2 :=
sorry
'''

### Natural Language Reasoning
- **Hypothesis 1**: The function 'f' is holomorphic in the open set 'Ω',
    represented by an assumption that 'f' is differentiable at every
    point 'z' in 'Ω'.
- **Hypothesis 2**: The real part of 'f' is constant on 'Ω', represented
    by the existence of a real number 'c' such that for every point 'z'
    in 'Ω', the real part of 'f(z)' equals 'c'.
- **Conclusion**: The function 'f' itself is constant on the set 'Ω',
    which means for any two points 'z1' and 'z2' in 'Ω', 'f(z1)' equals
    'f(z2)'.

We use 'sorry' as a placeholder for the proof details.
```

The extracted formalization is

```
theorem holomorphic_const_real_part_is_const {f : ℂ → ℂ} {Ω : set ℂ}
  (hΩ : is_open Ω) (hf : ∀ z ∈ Ω, differentiable_at ℂ f z)
  (h_re_const : ∃ c : ℝ, ∀ z ∈ Ω, (f z).re = c) :
  ∀ z1 z2, z1 ∈ Ω → z2 ∈ Ω → f z1 = f z2 :=
sorry
```

And this is the ground truth formalization adapted from ProofNet dataset.

```
theorem exercise_1_13a {f : ℂ → ℂ} ( Ω : Set ℂ ) (a b : Ω) (h : IsOpen
    Ω)
  (hf : DifferentiableOn ℂ f Ω) (hc : ∃ (c : ℝ), ∀ z ∈ Ω, (f z).re =
      c) :
  f a = f b := sorry
```

We can see that GPT-4o does preserved the semantic meanings of the original theorem, but it can not memorize the correct identifiers or terminologies in Mathlib4. This limitation is less problematic in elementary mathematics, where concepts are relatively simple and limited in scope. But as a general-purpose large language model not specialized in formal math, this becomes a significant constraint in advanced mathematical contexts, which involve a vast array of specialized concepts and terminologies. Maybe this can be alleviated via prompt engineering or retrieval augmentation, but exploring these approaches falls outside the scope of our current research.

## A.2 Data curation

### A.2.1 Prompt template for problem extraction

```
You are a helpful assistant that extracts complete math problems from
    text and formats them into a JSON list.

Format requirements:
- Each problem must include symbol definitions
- JSON keys: "problem" (description) and "type" ("proof"/"ans")

Output format:
[
  {"problem": "...", "type": "..."},
  {"problem": "...", "type": "..."}
]
```

### A.2.2  *Prompt template for problem validation*

```
You are a mathematical problem validator. Your task is to check whether
    a given mathematical problem meets the following criteria:

1. The problem should not be a definition, or descriptive statement.
2. The problem must explicitly state all necessary conditions and
    clearly state the conclusion to be proven or solved.
3. All symbols mentioned in the problem must be clearly defined and
    explained, except for standard mathematical terms and symbols in the
    {category} field that are conventionally understood. If the problem
    relies on well-known theorems or concepts, they should be referenced
    explicitly, but their detailed definitions need not be repeated.

If the problem meets all the criteria, return 'true' for the 'valid'
    field. Otherwise, return 'false'. State your answer as
    $\boxed{true}$ or $\boxed{false}$ at the end of your response.

Now, validate the following mathematical problem:

{problem}

Please think step by step and provide a detailed explanation before
    giving your final answer.
```

## A.3  Details of uproof dataset

Here is more representative examples in our uproof dataset. We also list the data sources and composition of uproof in Table A.3.

> **Topology.** Show that in terms of the ambient space the property of connectedness of a set can be expressed as follows: A subset $E$ of a topological space $(X, \tau)$ is connected if and only if there is no pair of open (or closed) subsets $G_1, G_2$ of $X$ that are disjoint and such that $E \cap G_1 \neq \varnothing$, $E \cap G_2 \neq \varnothing$, and $E \subset G_1 \cup G_2$.

> **Analysis.** Prove that the metric space $R[a, b]$ of real-valued Riemann-integrable functions defined on the closed interval $[a, b]$ is not complete with respect to the integral metric $d(f, g) = \int_a^b |f - g|(x)\, dx$.

> **Algebra.** Let $G$ be a group of order 15. According to the Third Sylow Theorem, the number of its Sylow 3-subgroups divides 5 and is congruent to 1 modulo 3. Show that there is one Sylow 3-subgroup, say $H$, and it is a normal subgroup.

> **Analysis.** Show that on any metric space $(X, d)$ one can introduce a metric $d^-(x_1, x_2) = \frac{d(x_1, x_2)}{1 + d(x_1, x_2)}$ in which the distance between the points will be less than 1.

| Book | Author(s) | Problems |
|------|-----------|---------|
| Mathematical Analysis I | Vladimir A. Zorich | 440 |
| Mathematical Analysis II | Vladimir A. Zorich | 433 |
| Algebra | Micheal Artin | 599 |
| Algebraic Topology | Allen Hatcher | 464 |
| Basic Topology | M. A. Armstrong | 165 |
| Complex Analysis | Elias M. Stein, Rami Shakarchi | 265 |
| Abstract Algebra | David S. Dummit, Richard M. Foote | 1355 |
| Functional Analysis | Theo B¨uhler, Dietmar A. Salamon | 330 |
| Commutative Algebra | M. E. Atiyah, I. G. MacDonald | 176 |
| Linear Algebra | Gilbert Strang | 129 |
| Linear Algebra and Geometry | Igor R. Shafarevich·Alexey O. Remizov | 183 |
| Probability: Theory and Examples | Richard Durrett | 247 |
| Real Analysis | Elias M. Stein, Rami Shakarchi | 239 |
| Statistical Inference | George Casella, Roger L. Berger | 248 |

Table 8: The sources and composition of uproof dataset.

## A.4 Details in reward design

**Syntax Check**. We leverage Lean 4 compiler and Mathlib4 library to conduct syntax check for each formalization. Relevant open-source projects are listed in Table 9.

| Name | Link | Version |
|------|------|---------|
| Lean 4 | https://github.com/leanprover/lean4 | v4.15.0 |
| Mathlib4 | https://github.com/leanprover-community/mathlib4 | v4.15.0 |
| Cli | https://github.com/leanprover/lean4-cli | v4.15.0 |
| REPL | https://github.com/leanprover-community/repl.git | v4.15.0 |

Table 9: Libraries and versions envolved in SC process.

For each formalized theorem we will add some predefined headers and imports before it is sent to the verifier. We used the same settings as Xin et al. (2024b). Here is the actual environment provided for each theorem:

```
import Mathlib
import Aesop
set_option maxHeartbeats 0
open Topology
open BigOperators
open Nat
open Real
open Rat
```

**Consistency Check**. We require a LLM to conduct consistency check with the following prompt template across our experiments.

```
Here is a natural language math problem and a translation in formal
    language Lean 4. You need to carefully analyse these problems and
    figure out wether they are equivalent or not. These problems must
    have exactly the same conditions and conclusions , they should be
```

```
marked false if they violate any of these requirements. You should
    reply false if the given formal statement is empty or in a weird
    format.

**Natural Language Problem**

{nl_statement}

'''lean
{fl_statement}
'''

State your answer as $\\boxed{{true}}$ or $\\boxed{{false}}$ at the end
    of your response.
```

The answer will then be extracted from response of the LLM, and can not obtain the expected output from the LLM, the default evaluation will be "false". Here is the sampling parameters we used in CC:

- Temperature: 0.6
- Min P: 0.05
- Max Completion Tokens: 2048

### A.5 Detailed training settings

We adopted a lightweight framework, trl(von Werra et al., 2020), for all of our training process. The actual version we used is v0.15.2.

**Baseline Settings**. We train our baseline formalizers via vanilla supervised fine-tuning method on Lean Workbook dataset.(Ying et al., 2024) Here is the detailed training hyperparameters of baseline formalizers:

- Learning Rate: $2 \times 10^{-5}$
- Weight Decay: 0
- Precision: bf16
- Train Epoch: 2
- Training Devices: 6
- Per Device Train Batch Size: 1
- Gradient Accumulation Steps: 1
- Max Seq Length: 4096
- Optimizer: AdamW

**RL Settings**. We conducted RL training with the same prompt template for instruction-finetuned models.

```
Translate the statement of this math problem into a single theorem in
    formal language Lean4. Do not write any proof steps for this theorem
    or try to solve this problem, you should focus on the translation
    and simply use 'sorry' as a place holder of the detailed proof. For
    example, 1+1=2 is translated into '''lean\nexample: 1+1=2 :=
    sorry\n'''.

### Natural Language Problem
{nl_statement}
```

Then we use GRPO algorithm to optimize our formalizer with the following hyperparameters, other hyperparameters are kept the same as the default settings in trl library.

- Learning Rate: $1 \times 10^{-6}$
- Per Device Train Batch Size: 1
- Num Generations: 4
- Train Epoch: 3
- Max Completion Length: 2048
- Gradient Accumulation Steps: 1
- GRPO Beta: 0.0
- Precision: bf16

### A.6 Additional experiment results

To assess reward hack issues in FormaRL, we also tested some other LLMs on the evaluation of the same samples generated by our formalizer. Results are summarized in Table 10. Models fine-tuned via FormaRL still outperformed SFT based models across all benchmarks.

| Method | Dataset | Evaluation Models | | |
|---|---|---|---|---|
| | | Qwen2.5-7B-Instruct | GPT-4o | DeepSeek-V3 |
| (Liu et al., 2025) | ProofNet | 19.68% | 16.98% | 17.52% |
| FormaRL | ProofNet | 31.81% | 27.49% | 26.68% |
| (Liu et al., 2025) | uproof | 6.2 % | 4.7 % | 5.1 % |
| FormaRL | uproof | 8.6 % | 5.8 % | 5.4 % |

Table 10: Pass@1 accuracy of both SC and CC on ProofNet and uproof with different large language model as the backend of consistency check. Thees results are all based on DeepSeek-Math-7B.

Here are some other additional experiment results not mentioned in the main part. They are scattered experiments conducted under different settings, and their conclusions are essentially consistent with those in the main text.

| Method | Base Model | miniF2F | ProofNet |
|---|---|---|---|
| SFT, | DeepSeek-Math-7B-Instruct | 97.34% / 57.79% | 67.39% / 36.66% |
| FormaRL+R | RAutoformalizer | 96.52% / 62.30% | 81.40% / 53.64% |
| (Liu et al., 2025) | RAutoformalizer | 96.72% / 36.07% | 75.74% / 39.89% |

Table 11: The pass@8 accuracy of in-distribution autoformalization performance. Statistics are reported as "SC pass rate / final pass rate".

| Model | Dataset | SC Pass Rate | Final Pass Rate |
|---|---|---|---|
| Qwen2.5-7B-Instruct | miniF2F | 20.90% | 10.66% |
| Qwen2.5-Coder-7B-Instruct | miniF2F | 34.22% | 22.54% |
| Qwen2.5-7B-Instruct | ProofNet | 2.43% | 1.62% |
| Qwen2.5-Coder-7B-Instruct | ProofNet | 6.73% | 4.04% |

Table 12: More results of vanilla ICL of different models. Performance are evaluated at pass@1 metric.

