# OpenReview forum: "FormaRL: Enhancing Autoformalization with no Labeled Data"
_colmweb.org/COLM/2025/Conference — COLM 2025_

### Official Review · Reviewer_bE1K · 2025-04-18

**Rating:** 7
**Confidence:** 4
**Ethics Flag:** 1

**Summary:**

This paper proposes FormaRL, a framework that trains LLMs for autoformalization using the GRPO algorithm on unlabeled data sources such as miniF2F and ProofNet. The authors leverage two forms of reward: syntax correctness validated by Lean, and consistency checks by LLMs. Additionally, they introduce uproof, a new dataset derived from undergraduate-level textbooks, to support and evaluate autoformalization performance. Experiments show that FormaRL outperforms standard SFT baselines, even those trained on significantly larger labeled datasets.

The paper is well-written and includes concrete examples that help illustrate the key ideas. The proposed method is intuitive and well-motivated, and the constructed dataset is a valuable contribution. It is particularly encouraging that the method achieves strong performance using limited unlabeled data. However, the core method primarily builds upon existing GRPO techniques and previously proposed reward designs, which somewhat limits the novelty of the contribution. In addition, the evaluation relies exclusively on proxy metrics such as compilation rate, without any accompanying manual analysis; this leaves open questions about qualitative correctness and robustness. Despite these limitations, the work marks a meaningful advance toward scalable auto-formalization, and in my view constitutes a worthy submission.

**Questions To Authors:**

If I understand correctly, the reward signal in GRPO relies on LLM-based checking. However, this introduces a potential concern: the reward may not be entirely reliable and could be susceptible to reward hacking due to the inherent limitations of LLMs in rigorous evaluation. Can the authors suggest alternative strategies to mitigate this issue?

Furthermore, a recent related work [1] introduces symbolic consistency checking as a more rigorous alternative for evaluating the quality of autoformalization. I am curious how the LLM-based consistency checking compares to these symbolic approaches in terms of accuracy and robustness. Can the authors comment on whether LLM-based checking is competitive, or if it could be combined with symbolic methods for improved performance?

[1] Autoformalize Mathematical Statements by Symbolic Equivalence and Semantic Consistency, NeurIPS 2024.

**Reasons To Accept:**

The paper is well-motivated and clearly written, with illustrative examples that effectively convey the proposed approach. The introduction of the uproof dataset is a valuable contribution that could benefit the community. Moreover, the experimental results are compelling—showing that training with limited unlabeled data can outperform SFT models trained on significantly larger supervised datasets—which makes the approach appealing for scalable and data-efficient autoformalization.

**Reasons To Reject:**

* The method builds primarily on existing GRPO techniques, which may limit its novelty.
* The evaluation relies exclusively on proxy metrics (e.g., pass rate) without any manual review of the auto-formalized outputs, leaving open questions about robustness.
* The submission employs an incorrect paper template and should be updated to comply with the conference guidelines.

---

> ### Author Response · Authors · 2025-06-03
>
> Many thanks for your constructive comments. Responses to the aforementioned questions are as follows. We sincerely hope them to resolve the concerns.
>
> > If I understand correctly, the reward signal in GRPO relies on LLM-based checking. However, this introduces a potential concern: the reward may not be entirely reliable and could be susceptible to reward hacking due to the inherent limitations of LLMs in rigorous evaluation. Can the authors suggest alternative strategies to mitigate this issue?
>
> Reward hacking is one of our primary concern as well, although in our experiments we believe that in our experiments there is no evidence of reward hack. We have also done some manual reviews on the outputs of RAutoformalizer and Qwen2.5-Coder-7B-Instruct trained with FormaRL, to further ensure the quality of formalizations and eliminate reward hack. We then compared the consistency check pass rate after passing syntax check under manual review and LLM-based checks, the results are summarized as follows.
>
> |Metric|Manual|Qwen2.5-7B-Instruct|
> |:-:|:-:|:-:|
> |RAutoformalizer|25%|37.4%|
> |Qwen2.5-Coder-7B-Instruct (FormaRL)|38.8%|50.4%|
>
> We can see that manual review is typically more strict than LLM-based consistency checks, while the latter might allow some incorrect samples to be accepted. These results also indicates that there is no evidence of reward hack in our experiments. Models trained with FormaRL do not exhibit significant higher CC pass rate under LLM-based check than manual review. Compared with that of RAutoformalizer, this difference is reasonable.
>
> In fact, in our previous experiments, as shown in Table 6 in our paper, the best results were not achieved by training with the same model as evaluation for CC. And if we consider their CC pass rate after passing SC (by dividing the final pass rate by the SC pass rate), FormaRL_(w/ qwen) actually performs worse than FormaRL_(w/ deepseek) in passing CC check, even though these metrics were evaluated under qwen.
>
> Here are some possible explanations about this phenomenon. Unlike reward models that directly provide a score according to some specific patterns, LLM will reason before making its judgement, and thus is harder to be hacked. And the existence of syntax check via Lean compiler further limits the output format of the formalizers. Although in our experiments reward hacking may not be an issue, an more accurate evaluation method would also help achieve better performance (i.e., FormaRL_(w/ deepseek) can achieve higher performance than FormaRL_(w/ qwen)).
>
> > Furthermore, a recent related work [1] introduces symbolic consistency checking as a more rigorous alternative for evaluating the quality of autoformalization. I am curious how the LLM-based consistency checking compares to these symbolic approaches in terms of accuracy and robustness. Can the authors comment on whether LLM-based checking is competitive, or if it could be combined with symbolic methods for improved performance?
>
> LLM-based evaluation approaches can be competitive in certain cases. In another recent paper [1], researchers have done some experiments and compared different evaluation metrics for autoformalization. In their experiments, majority voting via DeepSeek-V2.5 have achieved an overall accuracy of 82.50% (without the help of syntax check), which only falls behind their newly proposed BEq metric. In our quality assessment, both the recall and specificity of LLM-based consistency checks are around 80%, while syntax check would help further filter out incorrect samples.
>
> Another advantage of LLM-based CC lies in obviating the requirement for annotated datasets, so that it can be applied on new datasets as a filtering method.
>
> Nevertheless, these symbolic methods do exhibit higher precision and accuracy than LLM-based checks, and the idea of symbolic equivalence is really innovative and inspiring. I believe that we can achieve improved performance by integrating them with FormaRL. And with the help of RL, these symbolic methods could be directly adopted to enhance autoformalization, rather than merely serve as an evaluation metric. We are glad that we are the first to explore RL for autoformalization, and we are also looking forward to further experiments in this area.
>
> [1] Q. Liu, X. Zheng, X. Lu, Q. Cao, and J. Yan, “Rethinking and improving autoformalization: towards a faithful metric and a Dependency Retrieval-based approach,” in The Thirteenth International Conference on Learning Representations, 2025. [Online]. Available: https://openreview.net/forum?id=hUb2At2DsQ

---

> > ### Comment · Reviewer_bE1K · 2025-06-04
> >
> > Thanks for addressing my concerns. I encourage the authors to include the manual evaluation results in the final version and consider adding more examples if possible. I've updated my rating accordingly.

---

> > > ### Author Response · Authors · 2025-06-06
> > >
> > > Thank you for your appreciation! We will certainly include more results and observations in the final version of this paper, and we are excited about the future advancement of autoformalization.

---

### Official Review · Reviewer_MZ5Q · 2025-05-05

**Rating:** 7
**Confidence:** 2
**Ethics Flag:** 1

**Summary:**

This paper targets the Autoformalization capability of LLMs. Autoformalization is a challenging problem of translating natural language mathematics into formal languages like Lean. The authors make 2 major contributions -

1. They introduce FormaRL, a reinforcement learning framework that improves autoformalization capability of LLMs without requiring labeled training data.

2. They also create a new dataset to test the autoformalization capability of  LLMs - uproof, curated from 14 classical mathematics textbooks spanning core undergraduate curricula, including mathematical analysis, linear algebra, abstract algebra, real analysis, complex analysis, functional analysis, topology, probability, statistics and commutative algebra.

FormalRL Description
--------------------------
FormalRL is a reinforcement learning approach towards training a model for Autoformalization. It combines two components for reward calculation - syntax checking via the Lean compiler to verify syntactic correctness followed by consistency checking via large language model to ensure semantic alignment between the natural language statement and its formalization. Using these rewards, they train a LLM via the GRPO algorithm to create a better LLM at autoformalization.

Training and Evaluation Strategy
-------------------------------------
For SFT baselines, they train on  Lean Workbook - variant of miniF2F and ProofNet. For FormalRL - they train on miniF2F and ProofNet directly. Evaluation is done on uproof for 3 scenarios - pass@1, pass@8 and pass@16

Models explored are InternLM2 (From RAutoformalizer), DeepSeek-Math-7B-Instruct and Qwen2.5-Coder-7B-Instruct.

Results
---------
1. DeepSeek-Math-7B-Instruct - 0.8-5.8% improvment on top of SFT for various pass rates.
2. Qwen2.5-Coder-7B-Instruct - 2.1 - 14% improvement on top of SFT for various pass rates.
3. InternLM2 - 1.3% - 5.7% improvement over RAutoformalizer (a retrieval based method from (Liu et al., 2025).

**Questions To Authors:**

1. Lines 226-229, you claim that FromalRL on top of SFT can help improve further, but results plateau. I dont see those results in the paper or appendix. FormalRL should complement both SFT and RFormalizer, and potentially improve upon them. Some discussion with results around this would be good to have.
2. Can you expand a bit on Table 10, A.6. What was the ablation trying to achieve?
3. I was a bit surprised to see the data efficiency of this method - 859 examples only. Are there any variance numbers available for the results in terms of hyper parameter sensitivity?
4. How well does this method generalize to other theorem solvers - Isabelle and Coq?

**Reasons To Accept:**

1. Results shows improvement over baseline methods (SFT) and previous state of the art - RAutoformalizer
2. Their method used only 859 examples (sum of miniF2F and ProofNet), while the SFT methods used 25K samples and RAutoformalizer used 243k samples. This shows significant improvement in data efficiency. I found this result quite surprising given RL techniques tend to require significant more training data

**Reasons To Reject:**

1. The major contribution of this paper is to bring Reinforcement Learning with Verifiable Rewards paradigm to Autoformalization and showing that this works. While the results are indeed encouraging and (to the best of my knowledge) are the first to do this, approaching this problem using the GRPO with RLVR formulation is an obvious direction to pursue.

2. The lack of scalability studies to larger models and more theorem solvers make it hard to gauge the generalizability of their approach.

---

> ### Author Response · Authors · 2025-06-03
>
> Many thanks for your detailed and thoughtful review! Here we will respond to each comment point by point. I hope this could resolve some of your concerns.
>
> > Lines 226-229, you claim that FromalRL on top of SFT can help improve further, but results plateau. I dont see those results in the paper or appendix. FormalRL should complement both SFT and RFormalizer, and potentially improve upon them. Some discussion with results around this would be good to have.
>
> Sorry for not providing a reference in this paragraph. In fact, the relevant experimental results are already summarized in Table 3, 5.2, along with our major experimental results. We can see that RAutoformalizer is an autoformalizer trained via large scale SFT, and after performing FormaRL on this model we also observed some performance gains. The performance of SFT baselines are also summarized in this table for comparison.
>
> > Can you expand a bit on Table 10, A.6. What was the ablation trying to achieve?
>
> Thank you for taking the time to carefully read the content in the appendix. In fact this table consists of some in-distribution evaluation metrics of the same base model trained with three different approaches (SFT on Lean Workbook, FormaRL, RAutoformalizer). Results showed that our SFT method could achieve comparable performance with RAutoformalizer, while FormaRL outperformed both. Since in-distribution performance is not our primary focus, we placed these results in the appendix instead of main part.
>
> > I was a bit surprised to see the data efficiency of this method - 859 examples only. Are there any variance numbers available for the results in terms of hyper parameter sensitivity?
>
> In our early experiments we have tried many hyper parameter settings, and this RL training tend to be stable as long as the learning rate is not too high. In fact, current reinforcement learning from verifiable rewards (RLVR) tend to be quite efficient in data requirement compared to previous SFT-based methods. Before the submission of this paper, LIMR [1] already successfully incentived strong reasoning ability of an LLM with around 1k samples. Recently some experiments even showed that training on a single example could be effective as well [2]. And even for industrial-grade Qwen3 series models, only approximately 4k mathematical problems were utilized in the reinforcement learning stage [3]. The data efficiency and stability of RLVR was widely recognized across many researches, and we have just verified this in autoformalization.
>
> > How well does this method generalize to other theorem solvers - Isabelle and Coq?
>
> I believe this method could naturally generalize to other theorem provers, since we only leverages Lean compiler to conduct syntax check without depending on any other specific features of Lean. This training recipe can be easily adopted for other formal languages as well.
>
> [1] X. Li, H. Zou, and P. Liu, “LIMR: Less is More for RL Scaling.” [Online]. Available: https://arxiv.org/abs/2502.11886
>
> [2] Y. Wang et al., “Reinforcement Learning for Reasoning in Large Language Models with One Training Example.” [Online]. Available: https://arxiv.org/abs/2504.20571
>
> [3] A. Yang et al., “Qwen3 Technical Report.” [Online]. Available: https://arxiv.org/abs/2505.09388

---

> > ### Comment · Reviewer_MZ5Q · 2025-06-04
> > **Thank you for the reply, no change in rating**
> >
> > I appreciate the reply from the author. However, given my limited knowledge in this sub domain. I would not be able to further push the rating.

---

> > > ### Author Response · Authors · 2025-06-06
> > >
> > > That's alright. Thank you again for your appreciation and detailed reviews for this work!

---

### Official Review · Reviewer_iRck · 2025-05-16

**Rating:** 5
**Confidence:** 5
**Ethics Flag:** 1

**Summary:**

This paper propose, FormaRL, a reinforcement-learning approach to auto-formalizing math theorems (but not proofs). Additionally, a new dataset, uproof, is collected by consolidating 14 classical undergraduate-level textbooks. The reward design consists of syntax check (using Lean4) and consistency check (using LLMs). The experimental evaluation shows that FormaRL outperforms supervised finetuning on  the new dataset, uproof. The ablation study points out some concern of using LLMs as the judge for semantic correctness of formalized theorems.

**Questions To Authors:**

The pros and cons of this work are clearly stated, which is great, and there are no specific questions to be clarified.

**Reasons To Accept:**

- Using reinforcement learning to fine tune LLMs for auto-formalizing math theorems is a novel idea.
- An interesting dataset of consolidating classic textbook examples is constructed.
- The evaluation over the newly collected dataset shows interesting improvement over the supervised fine-tuning (SFT)-based approach.

**Reasons To Reject:**

- The evaluation metric is somewhat flawed, which is particular concerning given the essential objective of auto-formalization. LLMs are not good at precise logical reasoning, without mentioning checking the semantic equivalence of formal theorems.
-  Using LLM's feedback in the training process is acceptable, but there shall be rigorous quality in the final evaluation. The lack of rigorous evaluation make the measured improvement less convincing. Whether it is a genuine progress or a slightly more complicated reward hacking remains unclear.

---

> ### Author Response · Authors · 2025-06-03
>
> Thank you for your insightful review!
>
> You have pointed out one of our most important problem in our experiments, which we have paid a lot of attention to as well. To finally assess the output quality of our formalizers and the alignment of our evaluation metric, we have done some manual reviews on 200 samples separately generated by RAutoformalizer and Qwen2.5-Coder-7B-Instruct trained with FormaRL. We evaluated the consistency check pass rate after passing syntax check under manual review and LLM-based checks, the results are summarized as follows.
>
> |Metric|Manual|Qwen2.5-7B-Instruct|
> |:-:|:-:|:-:|
> |RAutoformalizer|25%|37.4%|
> |Qwen2.5-Coder-7B-Instruct (FormaRL)|38.8%|50.4%|
>
> We can see that manual review is typically more strict than LLM-based consistency checks, while the latter might allow some incorrect samples to be accepted. Nevertheless, the LLM-based CC check still exhibits similar trend as manual review, a higher CC pass rate under Qwen2.5-7B-Instruct typically corresponds to higher actual pass rate. So it could be acceptable to used this as an evaluation metric.
>
> These results also indicates that there is no evidence of reward hack in our experiments. Models trained with FormaRL do not exhibit significant higher CC pass rate under LLM-based check than manual review. Compared to that of RAutoformalizer, this difference is reasonable.
>
> In fact, in our previous experiments, as shown in Table 6 in our paper, the best results were not achieved by training with the same model as evaluation for CC. And if we consider their CC pass rate after passing SC (by dividing the final pass rate by the SC pass rate), FormaRL_(w/ qwen) actually performs worse than FormaRL_(w/ deepseek) in passing CC check, even though these metrics were evaluated under qwen. These results already eliminated reward hack, and I think a manual review would provide a more direct evidence. We believe that FormaRL does achieved some progress in autoformalization.
>
> Nevertheless, the evaluation metric adopted in our experiments does contain some flaws. Both the recall and specificity of LLM-based CC check are only around 80%, SC tend to be more reliable and can help eliminate more incorrect samples. This might impact the evaluation accuracy, but this does not affect our key experimental findings. In fact, the evaluation metric of autoformalization is still an area under exploration. Recently many new approaches [1][2][3] are proposed to enable more accurate evaluation for this, and they showed some superior performance than vanilla LLM-based checks. FormaRL exhibits better data efficiency and performance than SFT-based methods, and we believe that integrate this RL based method with better evaluation metric will further enhance the performance of autoformalization.
>
> I hope this could resolve some of your concerns!
>
> [1] Q. Liu, X. Zheng, X. Lu, Q. Cao, and J. Yan, “Rethinking and improving autoformalization: towards a faithful metric and a Dependency Retrieval-based approach,” in The Thirteenth International Conference on Learning Representations, 2025. [Online]. Available: https://openreview.net/forum?id=hUb2At2DsQ
>
> [2] Z. Li et al., “Autoformalize Mathematical Statements by Symbolic Equivalence and Semantic Consistency.” [Online]. Available: https://arxiv.org/abs/2410.20936
>
> [3] J. Lu, Y. Wan, Y. Huang, J. Xiong, Z. Liu, and Z. Guo, “FormalAlign: Automated Alignment Evaluation for Autoformalization.” [Online]. Available: https://arxiv.org/abs/2410.10135

---

> > ### Comment · Reviewer_iRck · 2025-06-04
> >
> > Thanks for sharing the manual review study. I am not fully convinced that a small-scale manual review would address the fundamental design flaw, especially given that the core task hinges on mathematical rigor and correctness. While I do not strongly oppose this work, I encourage the community to prioritize validating the correctness of formalizations rather than improving with a flawed metric.

---

> > > ### Author Response · Authors · 2025-06-06
> > >
> > > That's alright, I fully understand your concerns about experimental results. Although a small scale manual review might be the best we can do at the moment, we are still looking forward to future experiments with FormaRL. As mentioned above, many new evaluation metrics have been proposed for autoformalization recently. They demonstrate many advantages such as higher precision and specificity than LLM-based evaluation. I believe that integrating FormaRL with them will further validate the effectiveness of FormaRL, and push forward the cutting-edge performance of autoformalization.
> > >
> > > Thank you again for your critical comments on this work!

---

### Official Review · Reviewer_okNw · 2025-05-26

**Rating:** 6
**Confidence:** 4
**Ethics Flag:** 1

**Summary:**

The paper FormaRL: Enhancing Autoformalization with no Labeled Data proposes a reinforcement learning (RL) framework, FormaRL, for autoformalizing natural language math problems into formal Lean code without requiring labeled data. Instead of using expensive human annotations, FormaRL combines syntax validation from the Lean compiler with semantic consistency checks using large language models to compute rewards. The method significantly improves the autoformalization performance of models like Qwen2.5-Coder-7B-Instruct, achieving up to 4–6× accuracy gains on standard benchmarks (e.g., ProofNet). The authors also introduce uproof, a new dataset of over 5,000 undergraduate-level proof problems, to evaluate out-of-distribution generalization. Experiments show that FormaRL outperforms supervised fine-tuning (SFT) baselines while using far less data.

**Questions To Authors:**

- Line 168: To assess recall, is a language model used to perform consistency checks between the generated autoformalizations and the ground-truth formal statements?
- Line 171: Could variations in the algorithm for generating incorrect translations—such as altering statements to different degrees—impact the estimation of specificity?

**Reasons To Accept:**

- The paper is well written, easy to follow, and featuring comprehensive experiments.
- RL for autoformalization is a novel and exciting topic to explore.

**Reasons To Reject:**

- It would be valuable to include more qualitative analysis, especially given that the main evaluation relies on an unlabeled dataset (uproof) processed by LLMs. Equivalent representations of Lean formalizations may not be reliably captured by either the consistency check (CC) or the syntax check (SC).
- A comparative discussion between uproof and ProofNet would also be helpful, as both datasets potentially contain overlapping undergraduate-level math problems.

---

> ### Author Response · Authors · 2025-06-03
>
> Thank you for your review! Below we respond to each comment point-by-point.
>
> > A comparative discussion between uproof and ProofNet would also be helpful, as both datasets potentially contain overlapping undergraduate-level math problems.
>
> uproof is a large unlabeled dataset covering a wide range of mathematical topics, while ProofNet is a small manual curated dataset sourced from 8 undergraduate-level textbooks and Putnum competition. There might be some overlapping math problems, but considering that ProofNet only consists 371 problems, this may not be a problem.
>
> > Line 168: To assess recall, is a language model used to perform consistency checks between the generated autoformalizations and the ground-truth formal statements?
>
> Yes, in fact we directly passed the problems in natural language in ProofNet or miniF2F dataset and its ground truth formalizations to the large language models. The pass rate of these ground truth formalizations is just the recall of consistency check with different models.
>
> > Line 171: Could variations in the algorithm for generating incorrect translations—such as altering statements to different degrees—impact the estimation of specificity?
>
> You are right. Unlike the assessment of recall where we can directly evaluate on the ground truth formalizations, the assessment of specificity tend to be less reliable. Different formalizers tend to make different mistakes in autoformalization, so it is hard to evaluate a global specificity of our system. However, the estimation of specificity already provides valuable guidance for our evaluation metric. To preserve specificity we only use pass@n for syntax check and always use pass@1 for consistency check.
>
> For futther quality assessment we have done some manual reviews on 200 samples separately generated by RAutoformalizer and Qwen2.5-Coder-7B-Instruct trained with FormaRL. The estimated specificity of these samples are separately 83.5% for RAutoformalizer and 81.0% for our formalizer. These results are a little bit better than our initial simulated experiments though.
>
> I hope this could resolve some of your concerns.

---

> > ### Comment · Reviewer_okNw · 2025-06-05
> >
> > Thanks for sharing the manual analysis of the sampled problems. It might be helpful to include more qualitative insights, so we can better understand the types of errors the proposed model tends to make and how they differ (qualitatively) from those of other models. That said, I’m not strongly opposed to the paper being published — I just think it could benefit from a bit more polish. I’ll maintain my current rating.

---

> > > ### Author Response · Authors · 2025-06-06
> > >
> > > Thank you again for your reply and constructive comments!
> > >
> > > In fact in our manual reviews, we do have found some drawbacks of Qwen2.5 with FormaRL. One critical drawback is the unawareness of some terminologies in formal advanced math, such as CW complex, some probability distributions or functions with bounded variance. A common characteristic among them is that they all fall outside the training data (miniF2F and ProofNet). For comparison, RAutoformalizer, which was fine-tuned on 243k examples from informalized Mathlib4, can accurately distinguish and use these terminologies in its response.
> > >
> > > Nevertheless, RAutoformalizer still exhibits lower SC and CC pass rate, as it is trained to fit the given examples instead of generating a correct formalization. Maybe expanding the scope of training examples in FormaRL will help solve the drawback mentioned above, but this will require more experiments and its performance may still be bounded by pre-training.
> > >
> > > We will add more results and observations from manual review to the future version of this paper.

---

### Decision · Program_Chairs · 2025-07-08

**Decision:**

Accept

**Comment:**

I recommend accepting this paper. FormaRL shows impressive efficiency by improving autoformalization accuracy 4-6x using only 859 unlabeled examples, outperforming methods trained on 25K-243K samples. The authors effectively addressed evaluation concerns through manual reviews confirming genuine improvements rather than reward hacking. This data-efficient RL approach, combined with a valuable new dataset for undergraduate-level proof problems, represents a meaningful contribution to formal verification research.